# Research on the Response of Arbuscular Mycorrhizae Fungi to Grape Growth Under High Temperature Stress

**DOI:** 10.3390/ijms26136165

**Published:** 2025-06-26

**Authors:** Panyu Jian, He Zhang, Xiaojun Xi, Xiangjing Yin, Pengpeng Sun, Qian Zha, Dejian Zhang

**Affiliations:** 1Research Institute of Forestry and Pomology/Shanghai Key Labs of Protected Horticultural Technology, Shanghai Academy of Agricultural Sciences, Shanghai 201403, China; jianpanyu0001@163.com (P.J.); xxj220401@126.com (X.X.); yinxiangjing@saas.sh.cn (X.Y.); sunpp97@163.com (P.S.); 2College of Horticulture and Gardening, Yangtze University, Jingzhou 434025, China; zhangdejian0553@126.com; 3College of Horticulture, Hebei Agricultural University, Baoding 071001, China; zhanghebau@163.com

**Keywords:** grapes, arbuscular mycorrhizal fungi, photosynthetic efficiency, antioxidant enzyme, high temperature stress

## Abstract

Arbuscular mycorrhizae fungi (AMF) plays an important role in plants’ response to environmental stress, and the main environmental stress encountered in grape production is high temperature stress. This study aims to inoculate *Funneliformis mosseae* (A type of AMF) on grapes and investigate their tolerance to high temperature stress after inoculation. The results showed that AMF could infect grape roots, and the mycorrhizal infection rate was 20.78%. After inoculation with AMF, the growth of grape plants was significantly better than that in the non-inoculation group. Compared with the uninoculated group, the net photosynthetic rate, transpiration rate and stomatal conductance were higher in the AMF group, and the intercellular CO_2_ concentration was lower. After high temperature treatment, there was no significant difference in the content of hydrogen peroxide (H_2_O_2_) in grape leaves between the two experimental groups at each time, and the activities of superoxide dismutase (SOD), peroxidase (POD), catalase (CAT) and other enzymes showed great differences, especially after high temperature treatment for 6 h. The activities of SOD, POD and CAT in AMF group were significantly higher than those in uninoculated group. The content of malondialdehyde (MDA) in grape leaves of the two experimental groups had no significant difference between 0 h and 3 h after high temperature treatment, and the MDA content in the AMF group was significantly lower than that in the uninoculated group after 6 h of high temperature treatment. The contents of soluble sugar and soluble protein in the AMF group were higher than those in the uninoculated group at all time periods, especially after 6 h of high temperature treatment. In addition, we found that *VvHSP70*, *VvHSP17.9*, *VvGLOS1*, *VvHSFA2* genes all responded to high temperature stress, but there was no significant difference between the AMF group and the uninoculated group. It can be seen from the above that AMF can significantly enhance the adaptability of grape plants to high temperature stress by improving photosynthetic efficiency, antioxidant enzyme activity, soluble sugar and soluble protein content, and reduce Malondialdehyde (MDA) content, which provides guidance and theoretical basis for grape production.

## 1. Introduction

The grape is a woody vine in the Vitis [1]. The most important application of grapes is wine production, followed by raisins and juice, and is one of the world’s important economic fruit crops [2]. With the improvement of people’s living standards, consumption level capacity and consumer demand are also gradually changing, China’s grape planting area is expanding. China’s market demand for table grapes is very large, and China is the world’s largest producer and consumer of table grapes, grape industry has become a pillar industry in many local agricultural production, in rural revitalization, precision poverty alleviation work plays an important role.

Arbuscular mycorrhizae fungi (AMF) are microorganisms that are ubiquitous in soil and widely distributed in nature [3]. Mycorrhizae are symbionts formed by mycorrhizal fungi and plant roots, and are divided into three types: ectomycorrhizae, endomycorrhizae, and endophytic mycorrhizae. AMF is a typical endophytic fungus that can mutualize with many plants. Many studies have shown that AMF inoculation can improve the yield and quality of host plants and alleviate the damage caused by various adversities [4]. *Funneliformis mosseae* is a class of arbuscular mycorrhizal fungi widely distributed in soil ecosystems. It can substantially enhances plant growth by improving nutrient uptake and photosynthetic capacity [5]. AMF has been widely developed and applied in agricultural production [6].

High temperature stress means that the temperature rises to a certain extent and stays for a certain period of time, causing irreversible damage to crop production and development. Due to the frequent occurrence of global warming and extremely high temperature weather, environmental high temperature phenomenon has inevitably become one of the factors restricting agricultural crop production [7]. If plants are subjected to high temperature stress for a long time, they will produce excessive peroxides and damage cell membranes, resulting in the deterioration of the stability of membrane lipid peroxidation biofilms [8]. Degradation of nucleic acids, proteins and other macromolecules [9]; It also causes DNA double strand splitting; Enzyme activity decreased; The efficiency of enzymatic reaction decreased; Mitochondrial and chloroplast structures are damaged. The photosynthetic rate decreased significantly, and the plants were short and thin or stopped growing and died completely, which eventually led to a serious decline in plant yield and product quality [10]. How to effectively reduce the adverse effects of high temperature stress on crops and effectively ensure that crops can grow normally under high temperature stress has always been the focus of research workers.

Temperature is an important environmental factor affecting the growth and development of grapes [11], and high temperature stress will inhibit the growth and development of grapes and reduce the yield and quality of grapes [12,13,14,15,16]. High temperature during grape production can lead to obvious burning of grape leaves, abnormal softening of fruit and short fruiting period. The studies have proved that the optimal growth temperature of grapes is 25–30 °C [17,18]. High temperatures also affect the expression of relevant genes in grapes. In the study of grape leaves, Zha et al. [19] found that the antioxidant oxidase activity of grape leaves responds to high temperature stress, and high temperature induces the up-regulated expression of *VvGLOS1*, *VvHSFA2*, *VvHSP70*, *VvHSP17.9* and other genes, and the higher the temperature, the higher the gene expression. These results indicate that grape leaves can adapt to high temperature by regulating their own response mechanism through gene expression.

It can be seen from the above that frequent high temperature adversity has a deep impact on grape production, and how to enhance grape high temperature adaptability has a crucial role in grape production and development. Previous studies have shown that AMF can alleviate plant biological and abiotic stresses. Based on this, this paper explores the related physiological mechanisms of AMF to improve grape growth and enhance grape high temperature tolerance from the perspectives of appearance, physiology and biochemistry, and gene expression, so as to provide guidance and theoretical basis for grape production.

## 2. Results and Analysis

### 2.1. Effects of AMF on Grape Growth

After AMF infected the vines, the growth potential of the vines was significantly improved, and the vines inoculated with AMF were significantly taller, sturdier and had more leaves than those untreated. As shown in Table 1, shoots length, shoots thickness, leaf area, above ground fresh weight and underground fresh weight of grapes inoculated with mycorrhiza significantly increased by 55.95%, 15.89%, 34.24%, 23.91% and 50.45% compared with those treated without mycorrhiza. These results indicated that AMF could promote the aboveground growth of grape plants.

### 2.2. Mycorrhizal Growth of Grape Roots

After high temperature treatment, grape roots and planting soil were collected, and the mycorrhiza infection rate, soil spore density and soil mycelia length were measured. Through observation, obvious mycorrhizal structures were observed in the roots of AMF inoculated grapes (Figure 1). In addition, according to statistics, as shown in Table 2, the mycorrhizal infection rate of AMF inoculated plants was 20.78%, the soil spore density was 6.3 spores/g soil, and the soil mycelium length was 6.56 cm/g. Mycorrhizal infection rate, soil spore density and soil mycelia length were detected in uninoculated grapes. This indicates that grapes can be infected by AMF.

### 2.3. Effects of AMF on Grape Root Growth

As shown in Table 3, the projected area, volume and average diameter of vines in AMF group had extremely significant differences compared with those in Control group, increasing by 4.22%, 26.16% and 75%, respectively. Total length and total surface area of the root system were significantly different, increasing by 11.11% and 11.32%. There was no significant difference in surface area. These results indicated that AMF could promote the growth of underground part of grape plant.

### 2.4. Effects of AMF on Grape Chlorophyll Index Under High Temperature Stress

As shown in Figure 2, there was no significant difference in chlorophyll index of grape leaves between the Control group and the AMF group at 0 h, indicating that AMF-infected grapes had no effect on chlorophyll index of grape leaves. Under high temperature treatment, the chlorophyll index of grape leaves in the Control group showed a trend of first decreasing and then recovering. Compared with 42 °C treatment for 0 h, the chlorophyll index of grape leaves in the control group decreased by 5.3% after 3 h and recovered to the original level after 6 h. There was no significant change in chlorophyll index of grape leaves in the AMF group under high temperature treatment. The results indicated that the chlorophyll index of grape leaves in AMF group was more stable under high temperature treatment.

### 2.5. Effects of AMF on Photosynthetic Characteristics of Grape Under High Temperature Stress

As shown in Figure 3, there were significant differences in net photosynthetic rate, transpiration rate, stomatal conductance, and intercellular CO_2_ concentration of grape leaves between the Control group and the AMF group after 0 h of high temperature treatment, indicating that AMF infected grapes had an impact on photosynthetic characteristics of grapes. Under high temperature treatment, the net photosynthetic rate, transpiration rate and stomatal conductance of grape leaves in Control group and AMF group showed a decreasing trend. Compared with 0 h at 42 °C, the net photosynthetic rate, transpiration rate and stomatal conductance of grapes in the Control group decreased by 33.77%, 23.33% and 26.32%, respectively, after 3 h. After 3 h of high temperature treatment, the net photosynthetic rate, transpiration rate and stomatal conductance of AMF group decreased by 28.42%, 19.19% and 22.73%, respectively. Compared with the high temperature treatment for 3 h, the net photosynthetic rate, transpiration rate and stomatal conductance of grapes in the Control group for 6 h were decreased by 21.57%, 41.53% and 57.14%, respectively. The net photosynthetic rate, transpiration rate and stomatal conductance of AMF group decreased by 25%, 37.8% and 52.94%, respectively. In response to high temperature stress, the intercellular CO_2_ concentration of grape leaves in both treatment groups showed a trend of first increasing and then recovering to the original level. Compared with treatment at 42 °C for 0 h, the intercellular CO_2_ concentration of grape leaves in the Control group and AMF group increased by 3.19% and 3.58% respectively after 3 h, and recovered to the original level after 6 h. AMF inoculation could improve the net photosynthetic rate, transpiration rate and stomatal conductance of grape leaves under high temperature stress, and reduce the intercellular CO_2_ concentration, indicating that AMF enhanced the photosynthetic characteristics of grape plants.

### 2.6. Effects of AMF on MDA Content in Grape Leaves Under High Temperature Stress

As shown in Figure 4, there was no significant difference in MDA content of grape leaves between Control and AMF group after 0 h of high temperature treatment, indicating that AMF inoculation had no effect on MDA content of grape leaves. he MDA content of grape leaves in the two treatment groups showed a trend of continuous increase under high temperature treatment. Compared with 0 h, the MDA content of grape leaves in the Control group increased by 19.06% after high temperature treatment at 42 °C for 3 h, and 52.28% after high temperature treatment for 6 h. Under high temperature treatment, MDA content of grape leaves in AMF group increased by 20.34% at 3 h and 45.98% at 6 h compared with 0 h. These results indicated that AMF inoculation could reduce MDA content in grape leaves under high temperature stress.

### 2.7. Effects of AMF on H_2_O_2_ and Antioxidant Enzyme Activities of Grape Leaves Under High Temperature Stress

As shown in Figure 5, there was no significant difference in H_2_O_2_ content, SOD, POD and CAT activities in grape leaves between Control and AMF group after 0 h of high temperature treatment, indicating that AMF inoculation had no effect on H_2_O_2_ content, SOD, POD and CAT activities in grape leaves. Under high temperature treatment, the H_2_O_2_ content of grape leaves in the two treatment groups showed a trend of first increasing and then decreasing. Compared with 0 h, the H_2_O_2_ content of grape leaves in the Control group increased by 207.07% after 3 h high temperature treatment at 42 °C. After continued high temperature treatment for 3 h, it decreased to 166% of the original level. Compared with 0 h, the H_2_O_2_ content of grape leaves in AMF group increased by 168.35% after 3 h of high temperature treatment. It decreased to 134.36% of the original level after 3 h. These results indicated that AMF inoculation had no effect on H_2_O_2_ content in grape leaves under high temperature stress. SOD, POD and CAT activities of grape leaves in the two treatment groups showed an increasing trend under high temperature stress. Compared with 0 h, SOD and POD activities of grape leaves in Control group and AMF group were not significantly increased after 3 h of high temperature treatment. After 6 h of high temperature treatment, SOD and POD activities of grape leaves in Control group increased by 125.99% and 62.72%, respectively, compared with the original level. SOD and POD activities of grape leaves in AMF group increased by 119.41% and 77.91%, respectively. The increase rate of CAT activity in grape leaves of two treatment groups was slower than that of SOD and POD activity. Compared with 0 h, CAT activity of grape leaves in Control group increased by 49.71% after 3 h of high temperature treatment, and increased to 84.01% of the original level after 6 h of high temperature treatment. Compared with 0 h, CAT activity of grape leaves in AMF group increased by 62.52% after 3 h of high temperature treatment and 119.89% after 6 h of high temperature treatment. There was no significant difference in SOD, POD and CAT activities of grape leaves between the two treatment groups at 0 h, and significantly difference in SOD, POD and CAT activities of grape leaves between the two treatment groups after 6 h of high temperature treatment, indicating that AMF inoculation can improve SOD, POD and CAT activities of grape leaves under high temperature stress.

### 2.8. Effects of AMF on the Content of Osmotic Regulatory Substances in Grape Leaves Under High Temperature Stress

As shown in Figure 6, there was no significant difference in the contents of soluble sugar and soluble protein in grape leaves between the Control group and the AMF group after 0 h of high temperature treatment, indicating that AMF inoculation had no effect on the contents of soluble sugar and soluble protein in grape leaves. Compared with 0 h, there was no significant change in soluble sugar content of grape leaves in Ccontrol group after treatment at 42 °C for 3 h and 6 h. After 3 h of high temperature treatment, the soluble sugar content of grape leaves in AMF group was increased by 21.81% compared with 0 h. After high temperature treatment for 6 h, compared with the original level, the increase was 17.92%. These results indicated that AMF inoculation could increase the soluble sugar content of grape leaves under high temperature stress. Compared with 0 h under high temperature treatment, soluble protein content of grape leaves in the Control group increased by 50% after 3 h, and did not change much after high temperature treatment for 3 h, only increased by 32.14% of the original level. There was no significant difference in soluble protein content between AMF group and AMF group after 3 h of high temperature treatment. After high temperature treatment for 6 h, the increase was 97.3% compared with 0 h. These results indicated that AMF inoculation could increase soluble protein content in grape leaves under high temperature stress.

### 2.9. Effect of AMF on the Relative Expression of High Temperature Resistance Genes in Grape Under High Temperature Stress

In order to explore the effects of AMF inoculation on the expression of grape related genes under high temperature stress, grape high temperature response genes *VvHSP70*, *VvHSP17.9*, *VvGLOS1* and *VvHSFA2* were selected as objects, and the results were shown in Figure 7. Under high temperature stress of 42 °C, the expressions of four genes in both treatment groups were up-regulated. The expression patterns of the four genes varied at different times. *VvHSP70* gene in Control group was up-regulated from 0 h to 6 h after high temperature treatment, while *VvHSP70* gene in AMF group was up-regulated from 0 h to 3 h and down-regulated from 3 h to 6 h after high temperature treatment. There was no difference in *VvHSP70* gene relative expression between the two treatment groups after 0 h of high temperature treatment. The AMF group was significantly higher than the Control group after 3 h, and the AMF group was significantly lower than the Control group after 6 h. *VvHSP17.9*, *VvGLOS1* and *VvHSFA2* in the Control group all showed a trend of first increasing and then decreasing from 0 h to 6 h after high temperature treatment. In AMF group, only *VvHSFA2* and *VvHSP70* genes are consistent in expression. *VvHSP17.9* becomes stable after up-regulated expression from 0 h to 3 h after high temperature treatment, while *VvGLOS1* keeps rising. The relative expression of *VvHSP17.9* and *VvGLOS1* gene in AMF group and Control group was significantly higher than that in AMF group after 3 h of high temperature treatment, and *VvGLOS1* gene expression in AMF group was significantly higher than that in Control group after 6 h of high temperature treatment. The overall analysis showed that, under high temperature stress, no matter whether AMF was inoculated or not, the key genes of grape heat resistance were significantly responsive to high temperature. *VvHSP70* was regulated by AMF during high temperature treatment for 3 h, and *VvGLOS1* was regulated by AMF during high temperature treatment for 6 h. The other two genes did not show that they were regulated by AMF to alleviate high temperature stress in grape plants.

## 3. Discussion

This study showed that after AMF inoculation, mycelium structure could be observed in grape roots, and mycorrhizal infection rate, soil mycelium length and soil spore density were detected, indicating that AMF could symbiosis with grape roots. In addition, the results showed that the new shoots length, leaf area and fresh weight of AMF inoculated grapes were significantly higher than those of uninoculated grapes. The root growth in the underground part was also much better than that without inoculation, and the root length was longer, the diameter was thicker, the volume was larger, and the weight was heavier, indicating that AMF had a promoting effect on the growth and development of grape plants. After AMF infects grapes, it can expand the contact area between the grape roots and the soil, thereby enhancing the growth of grapes. This is consistent with the results of studies on corn [20], mung beans [21] and chili [22].

Photosynthesis is the main means for plants to obtain the energy needed for growth and development, and chlorophyll is indispensable for photosynthesis. The amount of chlorophyll contained in plants directly affects the strength of plant photosynthesis [23]. When plants synthesize chloroplasts, many enzymes are needed, and temperature is one of the main environmental factors affecting enzyme activity. Therefore, plant chlorophyll content often changes under high temperature stress. High temperature stress will reduce the synthesis of plant chlorophyll and accelerate its degradation [24,25], which will lead to wilt and yellowing of plants, which is seriously harmful to plants. A large number of studies have found that AMF inoculation can increase the chlorophyll content of plants under high temperature stress [26,27]. Mathur et al. [20] found that AMF inoculation under high temperature stress could increase the chlorophyll content of maize and enhance the tolerance of maize to high temperature. Matsubara et al. [28] found that the chlorophyll content of strawberries inoculated with AMF was significantly higher than that of uninoculated controls. Different from most experimental studies, this experiment only investigated the change of chlorophyll index of grape leaves inoculated with AMF within 6 h of high temperature treatment. The experiment found that The chlorophyll index of grape leaves in the Control group showed a phenomenon of first decreasing and then increasing. This might be because within the first three hours, the sudden high temperature led to a decrease in the synthesis rate of chlorophyll in grape leaves or an increase in the degradation rate, resulting in a decrease in the total amount of chlorophyll. After three hours of high temperature, the grapevines in the Control group have adapted to the high temperature. The dynamic balance of chlorophyll synthesis and degradation in leaves has been restored, and the total amount of chlorophyll has increased and returned to the normal level. The chlorophyll index of grape leaves inoculated with AMF was stable within 6 h of high temperature treatment, and did not decrease first and then increase as in the Control group, which proved that AMF inoculation could stabilize the chlorophyll index of grape plants under high temperature stress. This might be because inoculation with AMF can help grapes increase the synthesis rate of chlorophyll in leaves under high temperature stress or reduce the degradation rate, maintaining a dynamic balance between the synthesis and degradation of chlorophyll in grape leaves within six hours of high temperature stress, which is why the chlorophyll index remains stable.

Photosynthesis is the key to material conversion and energy metabolism in plant growth and development. The intensity of photosynthesis is also an important index to judge plant stress resistance [29]. Under high temperature stress, most plants will show a decrease in net photosynthetic rate [30]. This is due to the closure of some stomata and the decline of photosynthetic characteristics of mesophyll cells in plants subjected to high temperature stress [31]. In this experiment, the net photosynthetic rate, transpiration rate and stomatal conductance of grape plants were all decreased after high temperature stress. After high temperature treatment for 0 h, the net photosynthetic rate, transpiration rate and stomatal conductance of vines in AMF group were higher than those in Control group, and after high temperature treatment for 3 h and 6 h, the net photosynthetic rate, transpiration rate and stomatal conductance of vines in AMF group were still higher than those in Control group. The intercellular CO_2_ concentration in the AMF group was lower than that in the Control group at 0 h of high temperature treatment. With the increase of high temperature time, the intercellular CO_2_ concentration of the two groups increased first and then decreased to the original level, and the intercellular CO_2_ concentration of the AMF group was lower than that of the Control group at 3 h and 6 h. The experimental results showed that AMF inoculation could improve the photosynthetic characteristics of grape plants under normal environment and high temperature stress environment. It is indicated that inoculation of AMF under high temperature stress can reduce the damage to the chloroplast structure and photosynthetic organs of grape leaves and enhance the photosynthetic activity of mesophyll cells.

Plants sense changes in the environment through the plasma membrane of their cells. If plants are affected by high and low temperatures, waterlogging, salinity and drought, the cytoplasmic membrane is the primary site of damage. MDA is a product of lipid peroxidation of cell membranes, which can bind to soluble proteins and defensive enzymes, and thus damage the integrity of plant leaf cell membranes, resulting in loss of selective permeability and increase in cell conductivity [32]. When plants face high temperature stress, the membrane system is the first to be damaged, and the stability of the membrane can be used as an indicator of plant resistance to high temperature stress. MDA is the product of membrane lipidation, and its content can indirectly reflect the antioxidant capacity of plant tissues [33]. High temperature stress can break the dynamic balance of reactive oxygen production and clearance in plants, resulting in a substantial increase in MDA content [34]. A large number of previous experiments have proved that under high temperature stress, AMF inoculation can effectively reduce the content of MDA in plants, alleviate the degree of membrane peroxidation, and enhance the tolerance of plants to high temperature. Zhou et al. [35] research found that inoculation with AMF could significantly reduce the MDA content in tangerine leaves under high temperature and drought stress. Zhu et al. [36] found in their study of corn that under high temperature stress, the MDA content in AMF-inoculated and uninoculated plants increased, but the increase in mycorrhizal plants was lower than that in non-mycorrhizal plants. During the 6 h high temperature stress treatment in this experiment, during the 6 h high temperature stress treatment in this experiment, the increase rate and content of MDA in the AMF group were significantly lower than those in the control group. It indicates that AMF can effectively reduce the content of MDA in grapes under high temperature stress.

High temperature stress will cause a large amount of H_2_O_2_ accumulation in plants [37], resulting in oxidative stress on plants and inhibiting plant growth and photosynthesis [38]. Faced with this phenomenon, plants will reduce the harm caused by oxidative stress by increasing the activity of antioxidant enzymes such as SOD, POD and CAT in their bodies [39]. Li et al. [40] studied strawberries and found that SOD activity and high temperature tolerance of two kinds of strawberries were improved after inoculation with AMF. Shu et al. [41] also found that AMF inoculation under high temperature stress could increase the activities of SOD and POD in lily. This experiment only carried out high temperature stress for 6 h. Regardless of whether AMF was inoculated or not, the H_2_O_2_ content in grapes showed a trend of first increasing and then decreasing, and the difference between inoculation and non-inoculation was not significant. It was proved that high temperature stress could lead to the increase of H_2_O_2_ content in grapes within 6 h, and no matter whether AMF was inoculated or not, it did not significantly change its content and the rate of its increase. Combined with Figure 7, at 0 h of high temperature stress, the value of H_2_O_2_ content in the AMF group was slightly higher than that in the Control group, and significantly lower than that in the Control group at 3 h and 6 h. It is speculated that the reason for this phenomenon is the shorter high temperature time, resulting in the H_2_O_2_ content values between the two treatment groups not reaching a significant difference degree. However, since inoculation with AMF can help grapes remove H_2_O_2_, at all 3 h and 6 h, the H_2_O_2_ content values in the AMF inoculation group showed a lower phenomenon than those in the Control group. In the face of high temperature stress, the antioxidant enzyme CAT was the most sensitive, and the activity of CAT was detected at 3 h, while SOD and POD were detected at 6 h. Compared with no AMF inoculation, SOD, POD and CAT activities of high temperature treatment for 0 h and 3 h were not significantly different, and differences began to appear only after high temperature treatment for 6 h. The experimental results showed that AMF inoculation could improve the antioxidant oxidase activity of grape under high temperature environment. AMF eliminates excessive ROS and maintains the stability of cellular physiological functions by enhancing the activities of SOD, POD and CAT in grapes under high temperature stress. Since cell membrane lipid peroxidation involves the balance of multiple enzymatic and non-enzymatic as well as ROS protection systems, inoculation with AMF enhances the high temperature resistance of grapes by slowing down the lipid peroxidation process of grape cell membranes under high temperature stress, inducing the activity of antioxidant enzymes, and weakening cell membrane permeability.

High temperature stress can lead to the decrease of water content and water potential in plants, which seriously endangers plant growth and development [42]. Soluble sugar and soluble protein are important osmoregulatory substances in plants, which can help plants reduce osmosis, regulate osmotic pressure, and enhance high temperature resistance [43]. A large number of experiments have proved that AMF inoculation can effectively increase the content of soluble sugar and soluble protein in plants under high temperature stress. Xing et al. [44] found in their study of lilies that after inoculation with AMF under high temperature stress, the content of soluble protein in lily leaves significantly increased. Under high temperature stress, the soluble sugar content and soluble protein content of narrow-leaf lavender inoculated with AMF were significantly increased compared with those not inoculated [45]. In this experiment, with the increase of high temperature time, the contents of soluble sugar and soluble protein in grape leaves increased significantly, and the contents of both increased significantly in AMF plants than in uninoculated plants. These results indicated that AMF inoculation could effectively promote the synthesis of soluble sugar and soluble protein in plants under high temperature stress, and significantly reduce the damage of high temperature on grapes. 

When plants are stressed by adversity, they will turn on the self-protection mode, which is mainly manifested by the expression of stress genes [46]. By measuring the relative expression levels of four genes in grapes under high temperature stress, this study confirmed that *VvHSP70*, *VvHSP17.9*, *VvGLOS1* and *VvHSFA2* genes all respond after grapes are subjected to high temperature stress, and *VvHSP70* is regulated by AMF to relieve grape high temperature stress after 3h of high temperature stress. *VvGLOS1* was regulated by AMF to alleviate grape high temperature stress at 6 h of high temperature stress, while the other two genes did not show AMF regulation to alleviate grape high temperature stress at each time period. The four genes were selected based on previous studies, which showed that high temperature stress would affect the expression of these four genes in grapes [19]. This study only selected four genes because previous experiments have verified that these four genes are indicator genes for high temperature response. It is hoped that on this basis, further research can be conducted on whether AMF can regulate grape genes to alleviate high temperature? It is undeniable that the four genes cannot fully reflect the gene network by which AMF regulates the heat resistance of grapes. Although changes in the expression levels of grape genes after AMF inoculation can be detected, it is necessary to clarify which specific genes AMF can increase in expression? How is gene expression regulated? Further research is needed. In the future, through genomics combined with heat-resistant and non-heat-resistant germplasm resources, key genes can be explored to more comprehensively clarify the regulation of AMF on grape genes under high temperature stress.

## 4. Materials and Methods

### 4.1. Summary of Test Materials and Test Sites

The test strain was *Funneliformis mosseae*, and the test variety was self-cultivated annual grape ‘Shenfeng’ cutting seedling. The plastic pot with the upper diameter of 23 cm, the lower diameter of 20.5 cm and the height of 17 cm is used for planting. The soil was purified sand and autoclaved (121 °C, 0.1 MPa, 2 h) to eliminate indigenous fungal spores from the substrate.

The experiment site is located in the greenhouse of College of Horticulture and Landscape Architecture, Yangtze University, Jingzhou City, Hubei Province (112°08′52″ E, 30°21′11″ N). The average daytime temperature in the shed from March to June 2023 is 22.3 °C and the average night temperature is 12.7 °C. The highest temperature is 35 °C and the lowest temperature is 10 °C. Water and fertilizer management and pest control were carried out in 9 m × 3 m facility cultivation mode with conventional methods.

### 4.2. Experimental Design

The One year of growth cuttings of Shenfeng were 36 Pots with 1 plant per pot, among which 18 pots were treated with arbuscular mycorrhizal fungi (AMF) and 18 pots were not treated with inoculation (Control). The 36 pots of seedlings were divided into 3 groups (these three groups correspond to high temperature stress for 0 h, 3 h and 6 h respectively), with 12 pots in each group (6 pots for inoculation and 6 pots for non-inoculation). After growing at room temperature for trimester, the new shoot length, new shoot thickness and leaf area of grape were measured. Before the start of high temperature treatment, three groups of cuttings were darkened in a constant temperature incubator at 25 °C (13/11 h day/night, 85% RH, 0 LUX) for one night, and the constant temperature incubator was raised to 42 °C (13/11 h day/night, 85% RH, 10,000 LUX) at 9:00 am on the second day, and the chlorophyll and photosynthetic parameters of grape leaves were measured at 0 h, 3 h and 6 h of high temperature treatment, respectively, and randomly select 7 grape leaves from each group were quickly weighed and labeled. Cryostore at −80 °C, and the relative expressions of MDA, soluble sugar, soluble protein, H_2_O_2,_ SOD, POD, CAT and related genes were measured. After the high temperature was completed, soil samples were retained to measure the spore density, soil mycelium length, weight of above and below ground parts of plants, and the mycorrhiza infection rate and root configuration were measured by collecting and inoculating plant roots.

### 4.3. Test Methods

#### 4.3.1. Plant Growth and Development Status

The new slight length and leaf area were measured using Image J software (Version 1.8.0). The maximum diameter of the second section of the grape new shoot measured with a vernier caliper is the thickness of the new shoot. The fresh weight of the above-ground and underground parts of grapes was determined by using an electronic balance.

#### 4.3.2. Leaf Chlorophyll Index

The chlorophyll index was determined by portable plant polyphenol chlorophyll analyzer (Dualex Scientific+, Beijing Bopte Technology Co., LTD., Beijing, China).

#### 4.3.3. Leaf Photosynthetic Parameters

Leaf net photosynthetic rate, stomatal conductance and intercellular CO_2_ concentration were measured by Li-6400 photosynthesometer (LI-COR of the United States., Omaha, NE, USA). Before starting the experiment, the instrument cylinder should be in a ventilated and stable air pressure environment 2 m above the instrument main engine. The test was conducted every 3 h between 9:00 and 15:00, and 5 strains were randomly selected at each time point in each group. Finally, the data of each group were averaged. 

#### 4.3.4. Root System Configuration

WinRHIZO root analyzer (Hangzhou Hui ‘er Instrument and Equipment Co., LTD., Hangzhou, China) was used to obtain root configuration parameters (root length, surface area, projected area, volume, diameter, etc.).

#### 4.3.5. Mycorrhizal Growth

The research method of Phillips and Hayma [47] was used to determine the mycorrhizal infection rate by trimethylene blue staining. The mycorrhizal infection rate (%) = length of infected root segment/length of observed root segment × 100%.

The method of Bethlenfalvay and Ames [48] was used for microscopic examination under a biological microscope, and soil mycelia length and spore density were recorded and calculated.

#### 4.3.6. Determination of MDA Content in Leaves

MDA content detection kit AKF13C (Boxbio, Beijing Boxengong Technology Co., LTD. Beijing, China) was selected for MDA content determination.

#### 4.3.7. Determination of H_2_O_2_ and Antioxidant Enzyme Activities in Leaves

H_2_O_2_ content determination H_2_O_2_ content detection kit (AKO009C, Boxbio, Beijing BoxShenggong Technology Co., LTD., Beijing, China) was selected. SOD content detection kit (AKAO001C, Boxbio, Beijing Boxengong Technology Co., LTD., Beijing, China) was selected. POD content detection kit (AKAO005C, Boxbio, Beijing Box Shenggong Technology Co., LTD., Beijing, China) was selected. CAT content determination CAT content detection kit (AKEN001U, Boxbio, Beijing Box Shenggong Technology Co., LTD., Nanjing, China) was selected.

#### 4.3.8. Determination of Leaf Permeation Regulation Substances

Soluble sugar content detection kit (AKPL008C, Boxbio, Beijing Box Shenggong Technology Co., LTD., Beijing, China) was selected. Total protein assay kit (A045, Nanjing Jiancheng Bioengineering Research Institute Co., LTD., Neijing, China) was selected for the determination of soluble protein content.

#### 4.3.9. Analysis of High Temperature Related Gene Expression

RNA extraction was performed with plant total RNA extraction kit (Omega, Norcross, GA, USA), 1% agarose electrophoresis was used to detect the purity, and the RNA concentration was determined with NANO DROP2000 (Thermo Scientific, Waltham, MA, USA) and stored in a refrigerator at −80 °C for later use. 1 μg RNA was obtained and the first cDNA was synthesized by reverse transcription using PrimeScript II 1st Strand cDNA Synthesis Kit (TaKaRa, Bao Bioengineering Co., LTD., Dalian, China) for subsequent experiments. The relative expression levels of genes *VvGLOS1*, *VvHSFA2*, *VvHSP70* and *VvHSP17.9* were determined respectively. The internal reference gene selection *VvACTIN*, the selection of primers and the calculation of the relative expression levels of target genes were all referred to the methods of Zha et al. [18]. The setting of grape gene primers was derived from the NCBI database (Table 4).

#### 4.3.10. Data Analysis

The obtained data were expressed as Mean ± standard deviation (Mean ± SD), and the IBM SPSS Statistics data editor (Version 18.0) was used for significance analysis.

## 5. Conclusions

This experiment confirmed that compared with uninoculated grape seedlings, AMF inoculated grape seedlings grew at 25 °C for 3 months, and both the aboveground and underground growth potential were significantly enhanced, at 42 °C stable chlorophyll index, enhanced photosynthesis, increased antioxidant enzyme activity, and increased soluble sugar and soluble protein content. The expression of related high temperature response genes was also induced and regulated by AMF to varying degrees to cope with high temperature stress. Therefore, AMF can effectively enhance the resistance of grapes to high temperature stress through the above methods and reduce the damage caused by high temperature stress to grapes.

## Figures and Tables

**Figure 1 ijms-26-06165-f001:**
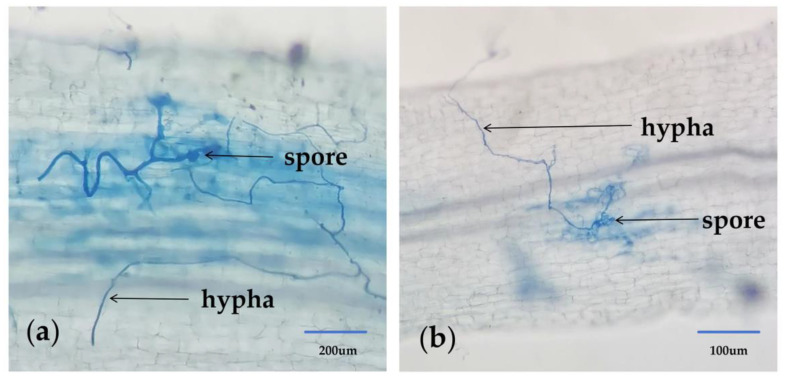
The situation of AMF infection in grape root system. Note: Both (**a**) and (**b**) are 10× observations under a microscope.

**Figure 2 ijms-26-06165-f002:**
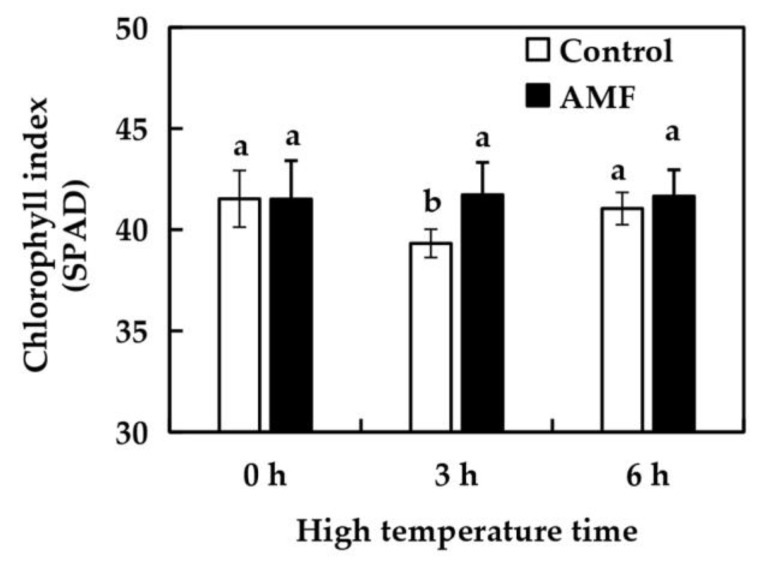
Effect of AMF on chlorophyll index of ‘Shenfeng’ leaves under high- temperature treatment. Control: not inoculated with AMF; AMF: inoculated with AMF; All data were analyzed by variance, and different letters indicated significant differences at the *p* < 0.05 level, the same as below.

**Figure 3 ijms-26-06165-f003:**
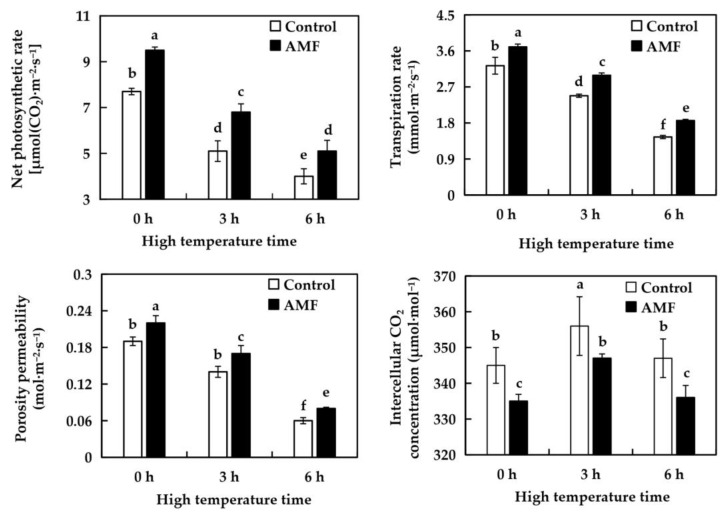
Effects of AMF on photosynthetic characteristics of ‘Shenfeng’ leaves under high-temperature stress. Control: not inoculated with AMF; AMF: inoculated with AMF.

**Figure 4 ijms-26-06165-f004:**
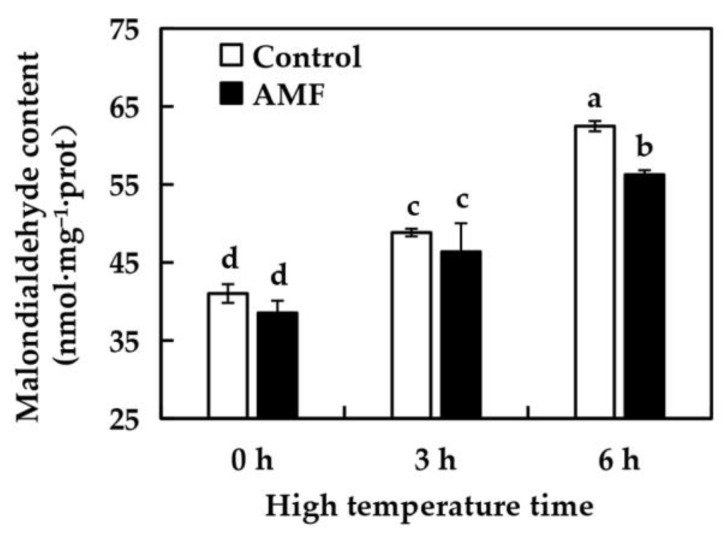
Effects of AMF on MDA content in ‘Shenfeng’ leaves under high-temperature stress. Control: not inoculated with AMF; AMF: inoculated with AMF.

**Figure 5 ijms-26-06165-f005:**
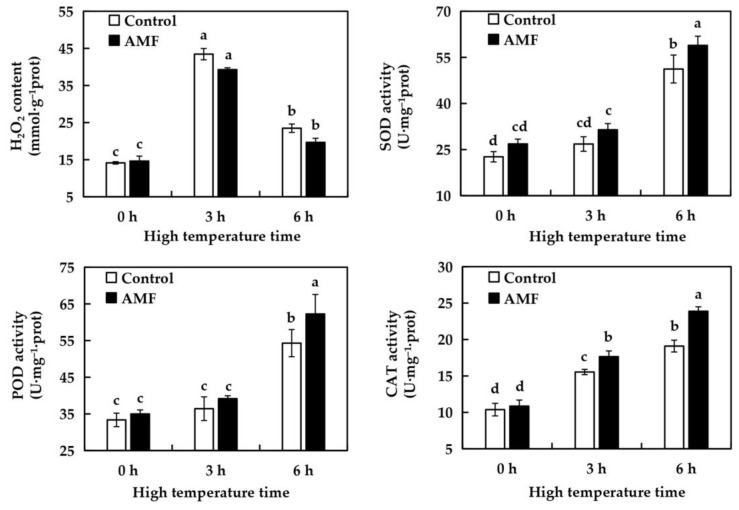
Effects of AMF on the activities of hydrogen peroxide and antioxidant enzymes in ‘Shenfeng’ leaves under different time of hig- temperature stress. Control: not inoculated with AMF; AMF: inoculated with AMF.

**Figure 6 ijms-26-06165-f006:**
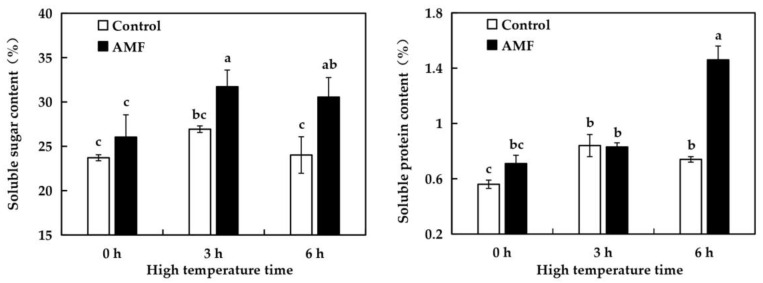
Effects of AMF on the contents of soluble sugar and soluble protein in ‘Shenfeng’ leaves under different time of high-temperature stress. Control: not inoculated with AMF; AMF: inoculated with AMF.

**Figure 7 ijms-26-06165-f007:**
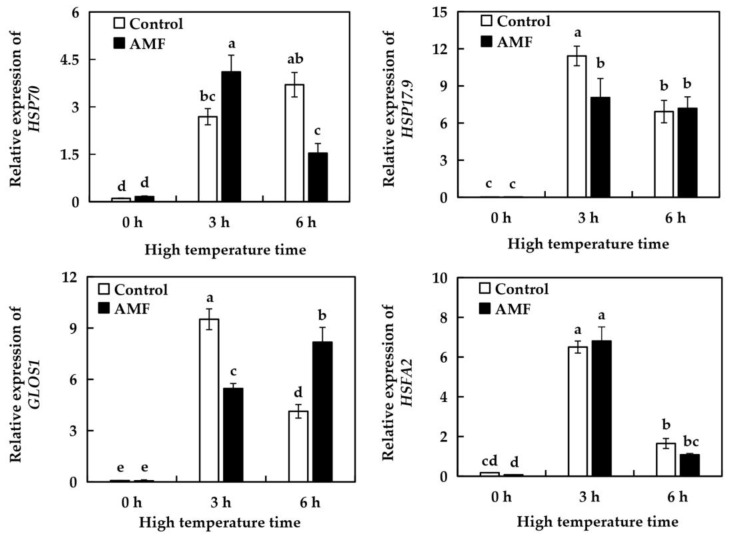
Influence of AMF on the relative expression levels of ‘Shenfeng’ genes *VvHSP70*, *VvHSP17.9*, *VvGLOS1* and *VvHSFA2* under high-temperature stress. Control: not inoculated with AMF; AMF: inoculated with AMF.

**Table 1 ijms-26-06165-t001:** Effect of AMF on the growth of Shenfeng.

Treatment	Shoot Length/cm	Shoot Thickness/mm	Leaf Area/cm^2^	Fresh Weight Above Ground/g	Fresh Underground Weight/g
Control	47.9 ± 4.7	4.3 ± 0.4	128.8 ± 12.5	32.2 ± 1.9	88.8 ± 2.7
AMF	74.7 ± 5.8 **	5.1 ± 0.2 **	172.9 ± 20.6 **	39.9 ± 1.7 **	133.6 ± 4.1 **

Note: The data in the table is the mean value; ±: standard error; Analysis of variance was performed on the same column of data, **: *p* < 0.01.

**Table 2 ijms-26-06165-t002:** Mycorrhizal growth of ‘Shenfeng’ root system.

Treatment	Mycorrhizal Infection Rate/%	Soil Spore Density/Spores·g^−1^	Soil Mycelium Length/cm·g^−1^
Control	-	-	-
AMF	20.78	6.3	6.56

Note: - Indicates no detection.

**Table 3 ijms-26-06165-t003:** Effects of AMF on root growth of ‘Shenfeng’.

Treatment	Total Root Length/cm	Projected Area/cm^2^	Surface Area/cm^2^	Volume/cm^3^	Mean Diameter/mm
Control	225.9 ± 19.3	14.2 ± 0.2	15.9 ± 0.6	22.9 ± 1.7	2.0 ± 0.2
AMF	251.0 ± 14.5 *	14.8 ± 0.3 **	17.7 ± 0.8 *	28.89 ± 3.3 **	3.5 ± 0.4 **

Note: Analysis of variance was performed on the same column of data, *: *p* < 0.05; **: *p* < 0.01.

**Table 4 ijms-26-06165-t004:** Real-time fluorescent quantitative PCR primer sequences of grape target genes.

Name	Sequence	NCBI No.
*VvGOLS1-qF*	TGATTACAGCAGCGTTTTGCC	VIT_07s0005g01970
*VvGOLS1*-qR	CGAGAGTACTGGCCTCTTCTAG	
*VvHSFA2-qF*	AGGCGGCTGGGACAATGAATC	VIT_04s0008g01110
*VvHSFA2-qR*	ATCCTCCACCTCCACATCAGTTTC	
*VvHSP70-qF*	CGGAGAAATGCGGCTGATA	TC38947
*VvHSP70- qR*	TCCCTTTACTTCCACCGCTAGA	
*VvHSP17.9-qF*	CGTCAAGGAGTACCCCAATTC	XM_002280644
*VvHSP17.9-qR*	AACTTCCCCACCCTCCTCT	
*VvACTIN-qF*	CTTGCATCCCTCAGCACCTT	EC969944
*VvACTIN-qR*	TCCTGTGGACAATGGATGGA	

## Data Availability

The data presented in this study are available within the article.

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
