# Peer review of "Research on the Response of Arbuscular Mycorrhizae Fungi to Grape Growth Under High Temperature Stress"

_ijms, 2025, doi:10.3390/ijms26136165_

Round 1
Reviewer 1 Report
Comments and Suggestions for Authors
This study investigates the role of arbuscular mycorrhizal fungi (AMF) in enhancing grapevine tolerance to high-temperature stress, with a focus on physiological indicators following AMF inoculation. The findings provide both theoretical and practical insights into plant stress responses.
The following issues require further discussion:
- Terminology clarification: The phrase "Glomus mosseae (AMF)" in the Abstract is scientifically inaccurate.
- Keywords: The term "gene" lacks relevance to the core methodology, whereas "antioxidant enzymes" should be added as a critical keyword.
- Lines 47-52 require citation of references to validate claims about AMF-mediated stress mitigation mechanisms.
- Given the availability of grape genome data, justification is needed for NCBI-based primer design rather than using genome-annotated sequences.
- Representative images of heat stress phenotypes (e.g., leaf wilting, chlorosis) must be included to support physiological data.
- The Introduction need add the mechanisms of AMF in thermotolerance and impacts on grapevine physiology.
- Provide experimental or literature-based evidence supporting the 42°C treatment protocol.
- Clarify whether Figure 3.1 includes photographic evidence of heat-induced morphological changes.
- Figure 1 requires explicit labeling of panels (a) and (b) in the caption and/or figure legend.
- The observed chlorophyll rebound in control groups at 6h (Figure 2) contradicts established patterns of heat-induced chlorophyll degradation.
- If root measurements in Section 3.5 were conducted under non- high temperature stress conditions, clarify treatment duration. Besides, is it appropriate to place it in this chapter?
- Discuss potential regulatory mechanisms of Hâ‚‚Oâ‚‚ trend in Section 3.7.
- In Section 3.9, provide evidence linking the four selected genes to AMF-mediated thermotolerance. It is suggested to select more candidate genes.
- The 0/3/6h sampling intervals lack justification. Why not measure more time points?
- Verify the validity of "6.3 #/g" (Line 195) as a quantitative unit for the described measurement.
- Provide histochemical (e.g., electrolyte leakage) or molecular evidence demonstrating that 6h exposure represents the critical threshold for heat injury in this cultivar.
Author Response
Dear reviewers,
Thank you very much for consideration of this manuscript and peer review with helpful suggestions. We have revised the manuscript according to your suggestions with red background that you can immediately recognize where the changes have been made. I replied to each of the questions you raised one by one in Word. Thank you again for your suggestions.

Reviewer 2 Report
Comments and Suggestions for Authors
In title AMF can be written in full form
Resarch gap, aims and scope in abstract are missing from abstract
Material method is mising in abstarct
Statistical analysis missing in abstract
Can you explain word unvaccinated used for plants.
Vitis is genus not family. Correct the sentence.
Make introduction in the transition between topics.
This study focuses on China’s grape industry, please incluse broader aspects as well.
It does not articulate clear research questions or hypotheses.
It is unclear how the 36 pots were grouped.
No mention of randomization or blocking to control for environmental variability.
Missing timeline: How long were plants grown before heat stress?
No description of soil composition
Missing light intensity, humidity, or photoperiod in the greenhouse.
How was the 42°C temperature maintained?
Why were 7 leaves weighed? Is this per plant or per treatment?
Fails to explicitly state whether differences between AMF and control groups at each time point (e.g., 3h, 6h) are statistically significant.
I am not satified with conclusion: Please revise it using these sections because it failed to connect results to broader implications
It does not explain why AMF improves thermotolerance
No acknowledgment of study constraints
Missing recommendations for future research.
Author Response

(The authors gave the same response as above.)

Reviewer 3 Report
Comments and Suggestions for Authors
The article is written correctly. It concerns the assessment of the effect of Glomus mosseae (AMF) on the growth of grapes and their response to high temperature (42℃). The course of photosynthesis, gas exchange indices of plants, hydrogen peroxide content and plant stress hormones were analyzed. The authors showed a positive effect of AMF on the activities of superoxide dismutase (SOD), peroxidase (POD), catalase (CAT) compared to the control group (not vaccinated). The effect of high temperature on some of the studied indices was also demonstrated. The introduction refers to the title of the work. However, there is no research hypothesis, which should be supplemented. The research methodology is correct, except for subchapter 2.3.1. Please describe how the research was performed in more detail (line 127). The results and discussion are described correctly. The authors cited the latest literature on the subject of the research. The conclusions refer to the aim of the work. The article can be published in IJMS after minor corrections.
Author Response

(The authors gave the same response as above.)

Round 2
Reviewer 1 Report
Comments and Suggestions for Authors
Your answers to the various questions I raised last time were too simple. Furthermore, you did not provide representative images of heat stress phenotypes, which leads me to think that there might be certain problems with this paper. Therefore, I believe this article cannot be published.
Author Response
Dear Editor and reviewers,
Thank you very much for consideration of this manuscript and peer review with helpful suggestions. We have revised the manuscript according to your suggestions with red background that you can immediately recognize where the changes have been made. I have organized the reply into a word document for your convenience to view.

Reviewer 2 Report
Comments and Suggestions for Authors
Dear author I would suggest to use curreent name of Glomus mosseae, which is Funneliformis mosseae.
Add study gap in second line of abstarct before methodology.
Also add some infomation background in introduction about Funneliformis mosseae.
As per my knowledge the sections of this mansucript are not in correct sequence. Introduction, material method, results, discussion conclusion should be followed.
MDA has not been used in full form.
Author cliams in line 439-444 that: A large number of previous experiments have proved that under high
temperature stress..... But author failed to provide references of previous experiments.
Author claim same in lines 483-485 but failed to provide related references.
Conclusion part is similar to abstract. revise it and add at least one novel finding from your results. Statistical results will be more suitable.
Author Response

(The authors gave the same response as above.)

Round 3
Reviewer 1 Report
Comments and Suggestions for Authors
The authors have addressed all my questions and concerns.
Reviewer 2 Report
Comments and Suggestions for Authors
Thanks for revision